# Deep Learning in the Biomedical Applications: Recent and Future Status

**Ryad Zemouri [1,*](ID), Noureddine Zerhouni [2] and Daniel Racoceanu [3,4]**

[1]  CEDRIC Laboratory of the Conservatoire National des Arts et métiers (CNAM), HESAM Université, 292, rue Saint-Martin, 750141 Paris CEDEX 03, France

[2]  FEMTO-ST, University of Bourgogne-Franche-Comté, 15B avenue des Montboucons, 25030 Besançon CEDEX, France; noureddine.zerhouni@femto-st.fr

[3]  Sorbonne University, 4 Place Jussieu, 75005 Paris, France; daniel.racoceanu@sorbonne-universite.fr

[4]  Scientific Director of Orqual Group, Kitview, 65 bd. Niels Bohr, 69100 Villeurbanne, France; daniel.racoceanu@kitview.com

*  Correspondence: ryad.zemouri@cnam.fr

**Abstract:** Deep neural networks represent, nowadays, the most effective machine learning technology in biomedical domain. In this domain, the different areas of interest concern the Omics (study of the genome—genomics—and proteins—transcriptomics, proteomics, and metabolomics), bioimaging (study of biological cell and tissue), medical imaging (study of the human organs by creating visual representations), BBMI (study of the brain and body machine interface) and public and medical health management (PmHM). This paper reviews the major deep learning concepts pertinent to such biomedical applications. Concise overviews are provided for the Omics and the BBMI. We end our analysis with a critical discussion, interpretation and relevant open challenges.

**Keywords:** deep neural networks; biomedical applications; Omics; medical imaging; brain and body machine interface

## 1. Introduction

The biomedical domain nourishes a very rich research field with many applications, medical specialties and associated pathologies. Some of these pathologies are very well-known and mastered by physicians, but others, much less. With the technological and scientific advances, the biomedical data used by the medical practitioners are very heterogeneous, such as a wide range of clinical analyses and parameters, biological parameters and medical imaging modalities. By the multitude of these data as well as the completeness of certain atypical diseases, biomedical data are usually imbalanced [1,2] and nonstationary [3], being characterized by a high complexity [1]. In this context, machine learning represents a tremendous opportunity: (1) to support physicians, biologists and medical authorities to exploit and significantly improve big medical data analysis; (2) to reduce the risk of medical errors; and (3) to generate a better harmonization of the diagnosis and prognosis protocols.

Artificial Neural Networks (ANNs) and Deep Learning (DL) are actually the leading machine-learning tools in several domains such as image analysis and fault diagnosis. The applications of the DL in the biomedical fields cover all the medical levels, starting from the genomic applications, such as the gene expression, to the public medical health management, such as predicting demographic information or infectious disease epidemics. Figure 1 outlines a rapid surge of interest in terms of the number of papers published in recent years in the biomedical applications, when deep learning is used. We can see that the number of research publications recorded an exponential growth during the last three years (comparing to other research field, e.g., fault diagnosis). The two main sub-fields,

namely medical/bioimaging and genomics, constitute the major part of these publications (around 70% total per year).

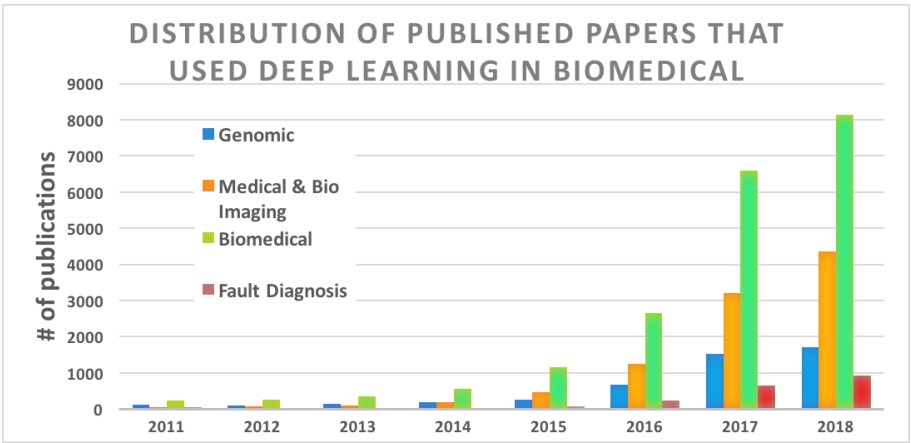

**Figure 1.** Distribution of published papers involving deep learning in biomedical applications. The statistics are obtained from Google Scholar; the search phrase is defined as the subfield name with deep learning, e.g., "genomics" and "deep learning".

The use of machine learning in biomedical applications can be structured into three main orientations: (1) as computer-aided diagnosis to help the physicians for an efficient and early diagnosis, with a better harmonization and less contradictory diagnosis; (2) to enhance the medical care of patients with better personalized therapies; and (3) to improve the human wellbeing, for example by analyzing the spread of disease and social behaviors in relation to environmental factors, or to implement a brain–machine interface for controlling a wheelchair [4]. To reach these three objectives, we segment the biomedical field into several sub-fields, as illustrated in Figure 2:

- Omics: genomics and proteomics are the study of the DNA/RNA sequencing and protein interactions and structure prediction.
- Bioimaging: the study of biological cell and tissue by analyzing the histopathology or immunohistochemically images.
- Medical imaging: the study of the human organs by analyzing magnetic resonance imaging (MRI), X-ray, etc.,
- Brain and body machine interface: the study of the brain and body decoding machine interface by analyzing different biosignals such as electroencephalogram (EEG), electroencephalogram (EEG), etc.
- Public and medical health management (PmHM): the study of big medical data to develop and enhance public health-care decisions for a humanity wellbeing.

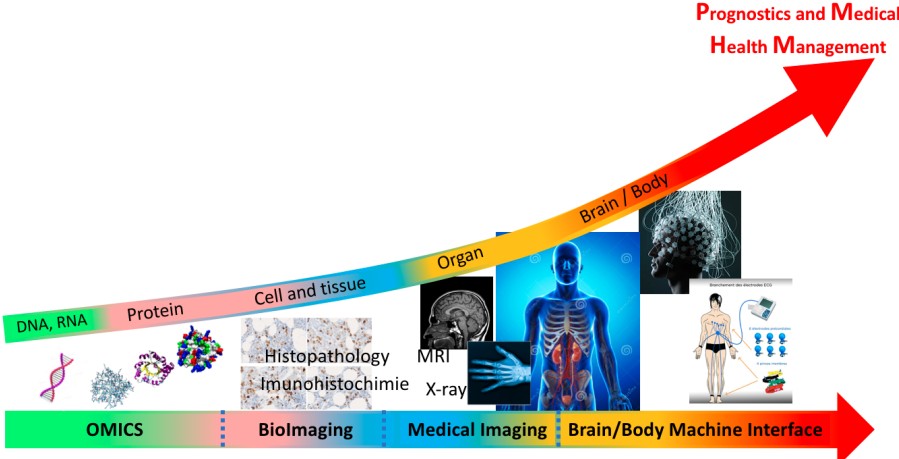

**Figure 2.** The different levels of the biomedical applications associated to each biomedical sub-field.

Several survey papers on the deep learning biomedical applications are recently published [5–10], where we can find all or part of the biomedical sub-fields, but with different appellations. Indeed, by analyzing these survey papers, we found a lack of harmonization for the definitions of certain sub-fields. Figure 3 outlines the contributions and the different points of view given in each survey paper. The biomedical research sub-field appellations used in this survey paper are given at the bottom of the figure. For example, the term "bioinformatics" used in [7] refers to the Omics, and the term "biomedical signal processing" used in [10] refers to brain and body machine interfaces (BBMIs).

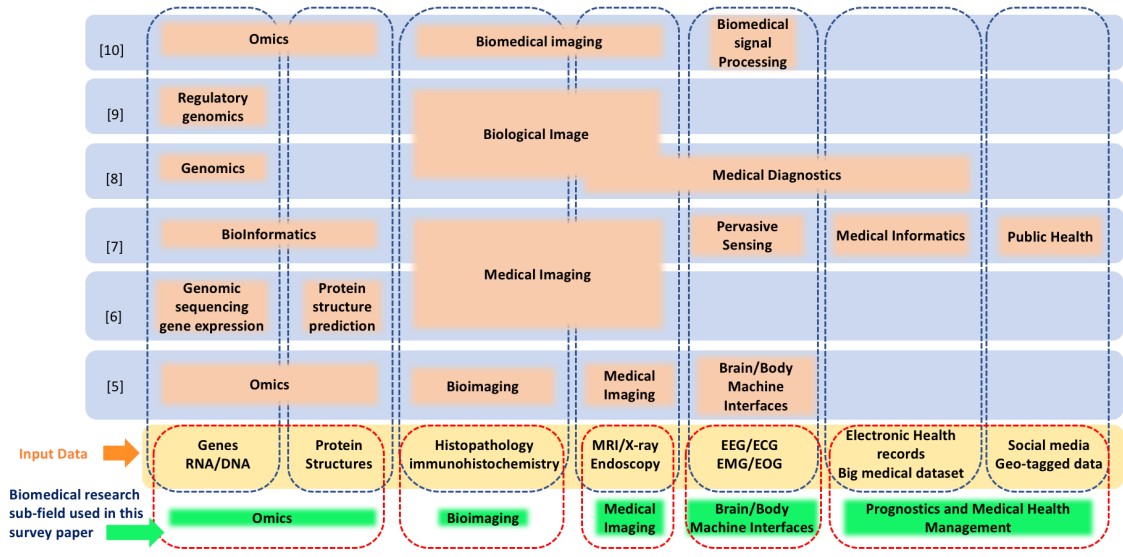

**Figure 3.** Different contributions and view points given by each survey paper recently published in the field of deep learning, applied to biomedical applications.

This survey paper is structured as followed. In Section 2, we introduce the main deep neural architectures that have been used for biomedical applications. Section 3 describes each field of the biomedical applications, namely the Omics, bio- and medical-imaging, brain and body machine interface (BBMI) and public medical health management (PmHM). We give in this section the main survey papers referring to the use of the deep neural networks. Sections 4 and 5 give a detailed descriptions of, respectively, the Omics and the BBMI with the main recent publications using the deep learning. Critical discussion and open challenges are given in Section 6.

## 2. Neural Network and Deep Learning

### 2.1. From Shallow to Deep Neural Networks

Artificial Neural Networks (ANNs) were inspired in the 1960s by biological neural networks in the brain [11,12]. The feed forward ANNs are composed by layers of interconnected units (neurons). The mathematical point of view of the ANNs consists of a non-linear transformation $y = F(x)$ of the input $x$ (Figure 4A). Compared to shallow architectures, ANNs with more hidden layers, called Deep Neural Networks (DNNs) [13], offer much higher capacity to learn fitting and feature extracting from high complexity input data (Figure 4B). The starting point of the deep learning was in 2006 with the greedy layer-wise unsupervised learning algorithm used for Deep Belief Networks (DBNs) [14–16].

The interconnection between two units or neurons has an associated connection weight $w_{ji}$, which is fitted during the learning phase. The input data are propagated from the input layer, neuron after neuron, until the output layer. This propagation transforms these data from a given space to another one, by the neurons of the layers in a nonlinear way. Each neuron computes a weighted sum of its inputs and applies a nonlinear activation function to calculate its output $f(x)$ (Figure 4C). The most used activation functions are: the sigmoid function and its variant the hyperbolic tangent for the shallow architectures, the rectified linear unit (ReLU) and its variant the softplus for the deep architectures, and the softmax commonly used for the final layer in classification tasks.

The two main applications of the ANNs are: *classification* and *regression*. The objective of the classification is to organize the input data space into several classes by supervised or unsupervised learning techniques. In the regression applications or function approximation, the objective is to predict an unknown output parameter—usually, by a supervised learning.

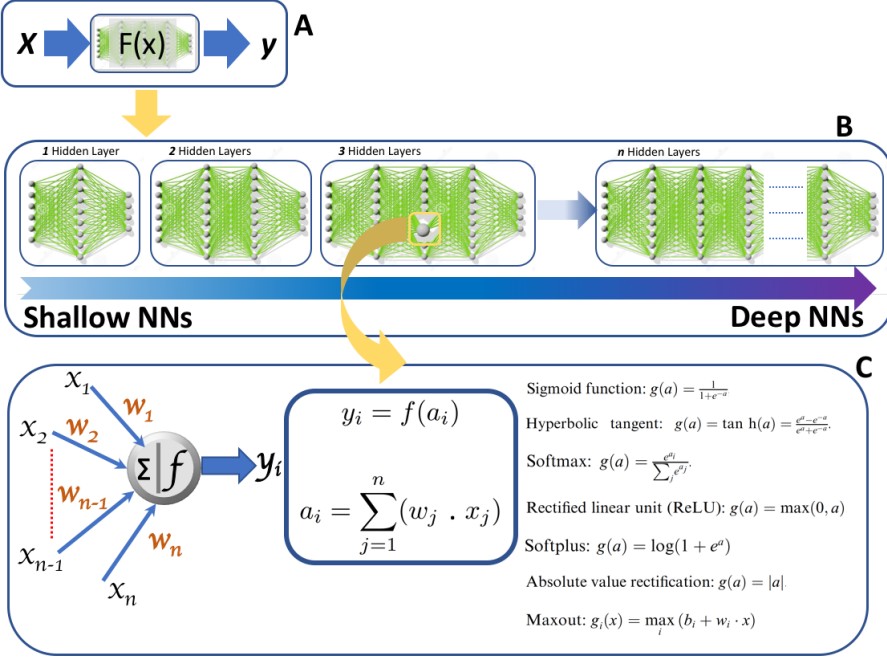

**Figure 4.** Artificial neural networks.

In supervised learning, the predicted label is compared with the true label for the current set of model weights $\theta_w$ to compute the output error (also called a loss function $L(\theta_w)$) (Figure 5A). This loss function is high-dimensional and non-convex with many local optimums. The learning phase consists of tuning the connection weights at each learning step to minimize $L(\theta_w)$ by backward propagating the gradient of the loss function through the network. This gradient backpropagation renewed ANNs in the mid-1980s when the backpropagation algorithm (BP) was used for classification [17]. During the learning procedure, two sets of data are usually used: training set and test set (sometimes a third set is

used for validation). The training set is used for the learning while the test set is used for the NN's performance evaluation. An efficient learning algorithm converges towards a global optimum while avoiding all the local optimums of the loss function, which looks like a landscape, with many hills and valleys [18]. A learning rate $\eta$ is used to jump over valleys at the beginning of the training and fine-tune the weights in later stages of the learning process. If the learning rate is too low (little jump), it may take forever to converge with a high risk of jamming in a local optimum. Conversely, a too high value (big jump) can cause a non-convergence of the learning algorithm (Figure 5B). Varying and adapting the learning rate during the training process produces better template update [19–26]. Another method grafting the stimulus sampling model onto the standard BP technique was also developed by Gorunescu and Belciug [27].

When deep architectures are used, the magnitude of the back propagated error derivative decreases rapidly along the layers, resulting in slight update of the weights in the first layers (Figure 5C). This drawback is partially solved by using the ReLU or Softplus activation function, which allows faster learning and superior performances compared to the conventional activation function (e.g., sigmoid or hyperbolic tangent) [28]. Another solution is to consider the learning rate as a hyper parameter where different learning rates are used for different layers but few works in the literature use this concept (see [29] for a review). The most popular method used to create deep architectures and solve the problem of the random initialization of the weight parameters is an unsupervised pre-training phase used before the supervised fined-tuned learning phase. Auto-Encoders (AEs) and Restricted Boltzmann Machine (RBM) are stacking in a layer-wise as the basic building blocks [13,30].

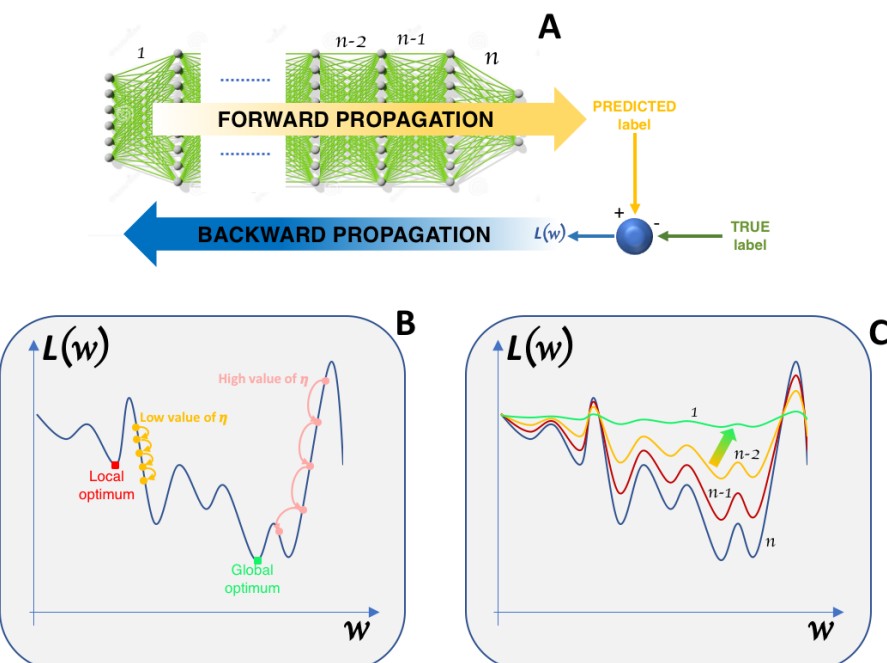

**Figure 5.** The Loss function and the backward propagating of its gradient. (**A**) The predicted label is compared with the true label for the current set of model weights $\theta_w$ to compute the output error (also called a loss function) $L(\theta_w)$. (**B**) The learning rate $\eta$ is used to jump over valleys. If the learning rate is too low (little jump), it may take forever to converge with a high risk of jamming in a local optimum. Conversely, a too high value (big jump) can cause a non-convergence of the learning algorithm. (**C**) For deep architectures, the magnitude of the back propagated error derivative decreases rapidly along the layers, resulting in slight update of the weights in the first layers.

Finding the best neural parameters set that minimizes the loss function is still challenging, especially for the DNN learning, which remains an active research area. Another difficulty encountered

during the learning process is the choice of the neural network architecture for a given problem in terms of number of hidden layers and units per layer. What are the criteria that define the number of hidden layers and neurons per layer? The greater is the learning base, the greater is the need to use deep architectures. The user often proceeds by trying several neural network topologies to find the best structure and try to avoid the oversized and undersized structure (Figure 6). This is a real drawback and computationally expensive, especially when deep architectures are used [13,29,31,32]. There is no guarantee that the selected number of hidden layers/units is optimal. To prevent the network from overtraining, caused by an oversized design, some regularization techniques are used, such as dropout [33,34], Maxout [35] or weight decay, which is a penalty added to the error function [36].

Evolutionary learning procedures [37] also give interesting solutions where the NN evolves gradually during the training procedure into an optimum structure that satisfies some evolutionary criteria. These adaptive neural networks are divided into three categories [38]: constructive or growing algorithms, pruning algorithms and hybrid methods.

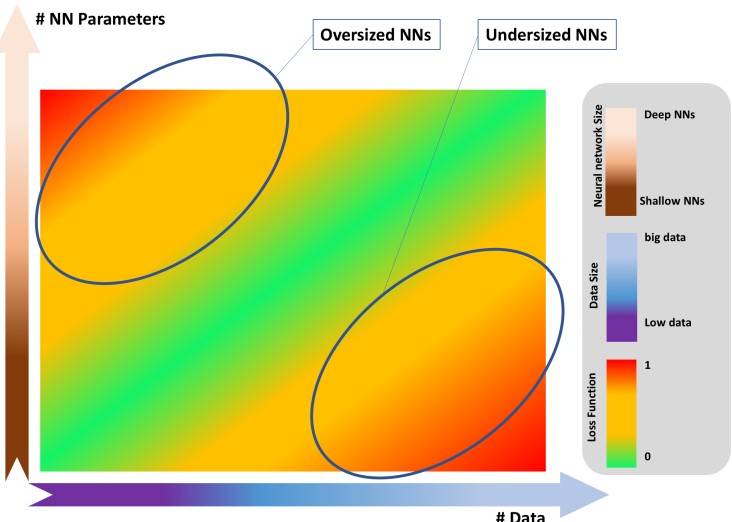

**Figure 6.** The correlation between the size of the training data and NN parameters.

### 2.2. Most Popular Deep Neural Networks Architectures Used in Biomedical Applications

In this review paper, we consider only feedforward NN architectures, namely ANNs without recurrences. Recurrent Neural Networks (RNNs) consider the dynamic aspect of the data by the existence of self-connections or connections between units of the same layer. Few biomedical applications use RNNs because of the complexity of their training process. However, the long short-term memory (LSTM) is the most used RNNs architecture in biomedical applications [7].

We also consider two kinds of data used in biomedical applications: data with and without local correlations. In the first case, the input data are represented by a one-dimensional (1D) vector, while, in the second case, the data are represented by a multidimensional matrix. In this case, there is a high local correlation (e.g., RGB images) in the input data, and it is very important to consider this local correlation during the learning process. In this section, we give the most used deep architectures for each type of input data: DNNs used for non-correlated data and DNNs used for high locally correlated data.

#### 2.2.1. DNN for Non-Locally Correlated Input data

For this type of data, three architectures are used: deep multilayer perceptron (DMLP), deep auto-encoders (DAEs) and deep belief network (DBN).

***Deep multilayer perceptron***

The well-known multilayer perceptron developed in 1986 by Rumelhart and trained by the backpropagation algorithm is the ancestor of deep learning [13,17] (Figure 7A–C). At that time, the maximum number of hidden layers was at least two or three, with few units per layer. Nowadays, due to the development of several heuristics for training large architectures and the use of GPUs hardware ([39–41]), the size of the NNs can reach several hidden layers with more than 650 thousand neurons and 630 million of trained parameters (e.g., alexNet [42]).

*Deep auto-encoders*

An auto-encoder is a special case of a one hidden layer MLP (Figure 7D). The aims of an AE is to recreate the input vector $\hat{x} = F(x)$ where $x$ and $\hat{x}$ are, respectively, the input and the output vector. In an AE, an equal number of units are used in the input/output layers and less units in the hidden layer. Deep AEs are obtained by stacking several AEs (Figure 7E). In deep learning, DAEs are used even for feature extraction/reduction, or to pre-train parameters for a complex network.

*Restricted Boltzmann machine and deep belief networks*

A restricted Boltzmann machine (RBM) is a generative stochastic network and consists of one layer of visible units and one layer of hidden units, with no connections within each layer [30,36,43] (Figure 7F). In other words, the visible and hidden layers correspond to the observation and the feature extractor, respectively. The resulting model is less general than a Boltzmann machine, but is still useful, for example it can learn to extract interesting features from images. RBM is considered as an energy based model where the energy of the joint configuration of visible and hidden layer units is defined as a Hopfield energy function [44]. This energy function assigns probability to each pair of visible and hidden vectors in the modeled network [45].

Traditionally, RBM is used to model the distribution of the input data or to model the joint distribution between the input data and the target classes [46]. In deep learning, similar to AEs, RBMs can also pre-train parameters for a complex network.

A deep belief network can be viewed as a stack of RBMs where the hidden states of each previous RBM are used as data for training a new second RBM [13,14,47,48] (Figure 7G). Therefore, each RBM perceives pattern representations from the level below and learns to encode them in unsupervised fashion.

2.2.2. DNN for High Locally Correlated Data

The only architecture used when there is a high local correlation within the data, is the Convolutional Neural Network architecture.

*Convolutional neural networks*

The convolutional neural networks (CNNs) were inspired by the neurobiological model of the visual cortex, where the cells are sensitive to small regions of the visual field [11,49–51]. CNNs are used to model input data in the form of multidimensional arrays, such as two-dimensional images with three colour channels. In CNN architectures, there are two types of layers: the convolutional layer and the pooling layer (Figure 7H).

A convolutional layer consists of multiple maps of neurons, so-called feature maps or filters. Unlike in a fully-connected network, each neuron within a feature map is only connected to a local patch of neurons in the previous layer, the so-called receptive field. The input data are then convolved by different convolutional filters via shifting the receptive fields step by step. The convolutional filters share the same parameters in every small portion of the image, largely reducing the number of hyper parameters in the model.

A pooling layer, taking advantage of the stationarity property of images, takes the mean, max, or other statistics of the features at various locations in the feature maps, thus reducing the variance and capturing essential features [6,9].

A CNN typically consists of multiple convolutional and pooling layers, which allow learning more and more abstract features. In the last layers of a CNN, a fully-connected classifier is used for the classification of the features extracted by the previous convolutional and pooling layers. The most popular CNNs used in the machine learning applications are: AlexNet [42], Clarifai [52], VGG [53], and GoogleNet [54].

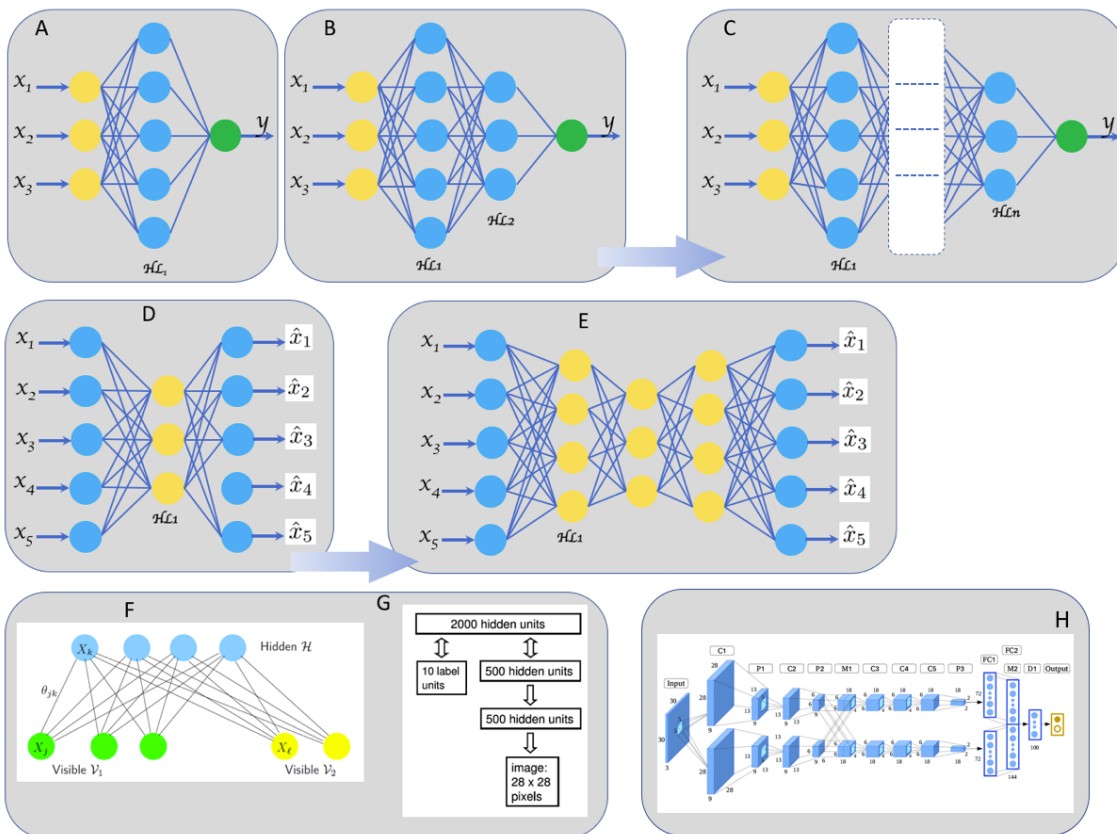

**Figure 7.** Most popular Neural Networks architectures used in biomedical: (**A**–**C**) feedforward neural networks with several depths (one hidden layer, two hidden layers and a deep architecture with many hidden layers); (**D**,**E**) an Auto-Encoder and a Deep Auto-Encoder, respectively; (**F**,**G**) representations of a restricted Boltzmann machine and a deep belief networks, respectively; and (**H**) an AlexNet [42] convolutional neural network.

## 3. Biomedical Applications

### 3.1. Omics

The first level of the biomedical research field concerns all studies starting from the genomic sequencing and the gene expression to the protein structure prediction and its interaction with other proteins or drugs. This is a very active research domain where the use of the deep neural networks is growing rapidly (see Figure 1). Usually, in the literature, the generic name for this research area is the "Omics" but other appellations can be found such as *bioinformatics* [7] or *biomedicine* [55] (see Figure 3 for the different research sub-field denominations found in the literature). The Omics covers data from genetics and (gen/transcript/epigen/prote/metabol/pharmacogen/multi) Omics [5] and aims to investigate and understand biological processes at a molecular level to predict and prevent diseases by involving patients in the development of more efficient and personalized treatment. A great

part of the Omics concerns protein–protein interactions (PPIs) [56], the prediction of human drug targets and their interactions [57], and the protein function prediction [58]. Table 1 gives the recent published survey papers on Omics. We recommend reading the papers by Leung et al. [59] and Mamoshina et al. [55], who provided a complete overview of genomics and major challenges in the machine learning practical problems.

**Table 1.** Survey papers on deep learning in Omics.

| Ref. | Year of Publi. | # of Citations | Research Topics |
| --- | --- | --- | --- |
| [60] | 2018 | 1 | Brief survey on machine learning used for the prediction of DNA-binding residues on protein surfaces |
| [55] | 2016 | 122 | Biomarkers, genOmics, transcriptOmics, proteOmics Structural biology and chemistry Drug discovery and repurposing Multiplatform data (MultiOmics) Challenges and limitations of deep learning systems |
| [59] | 2016 | 58 | A complete survey paper on how machine learning can be used to solve key problems in genomic |
| [61] | 2015 | 304 | Provides an overview of machine learning as it is applied to practical problems and major challenges in genOmics |

### 3.2. Bio and Medical Imaging

The next stage after the DNA and the protein level is the study of the cell (cytopathology) and the tissue (histopathology). Cytopathology and histopathology are commonly used in the diagnosis of cancer and some infectious diseases and other inflammatory conditions. The histological and cytopathological slides, usually obtained by fine-needle aspiration biopsies, are examined under a microscope [62,63]. This research field, called in the literature as bioimaging, is the main research area of deep learning in biomedical applications (see Figure 1).

In medical imaging, the human organs are studied by analyzing the different (medical/clinical/health) imaging [5]. Nowadays, large medical high-resolution imaging acquisition systems are available, such as parallel magnetic resonance imaging (MRI), multi-slice computed tomography (CT), ultrasound (US) transducer technology, digital positron emission tomography (PET) or 2D/3D X-ray. These medical images contain a wealth of information, causing some disagreement across interpreters [64]. The manufacturers of medical imaging systems try to provide software, workstations and solutions for archiving, visualizing and analyzing images [64].

In bio and medical imaging, the accurate of the diagnosis and/or assessment of a disease depends on both image *acquisition* and image *interpretation* [65]. Image acquisition has improved substantially over recent years with the development of the technology. Medical images interpretations is mostly performed by physicians and can be subject to large variations across interpreters, and fatigue (read the guest editorial [65]). Indeed, the main part of the deep learning applications in bio and medical imaging concerns the computer-aided images interpretations and analyses [66–70]. For example, analyzing histopathology images for breast cancer diagnosis [63,71–73], or a digital pathology and image analysis with a focus on research and biomarker discovery [74]). The major part of the deep learning papers published in bio and medical imaging concerns the segmentation, localization and classification of the nuclei [75] and mitosis [76] for bioimaging, and lesion and anatomical object (such as organ, landmarks and other substructures) for medical imaging. Table 2 provides the recent published survey papers in DL for medical and bioimaging. We recommend the work of Litjens et al. [77] for a complete overview.

**Table 2.** Survey papers on Deep Learning in bio and medical imaging.

| Ref. | Year of Publi. | # of Citations | Research Topics |
|---|---|---|---|
| [77] | 2017 | 563 | A complete survey on deep learning in bio and medical imaging:<br>- Image/exam classification<br>- Object or lesion classification<br>- Organ, region and landmark localization<br>- Organ and substructure segmentation<br>- Lesion segmentation<br>- Anatomical application areas (brain, eye, chest, digital pathology and microscopy, breast, cardiac, abdomen, musculoskeletal<br>- Anatomical/cell structures detection,<br>- Tissue segmentation,<br>- Computer aided disease diagnosis/prognosis |
| [78] | 2016 | 18 | ANNs as decision support Tools in cytopathology: past, present, and future:<br>- Gynecological cytopathology, the PAPNET System<br>- Cytopathology of gastrointestinal system<br>- Cytopathology of thyroid gland<br>- Cytopathology of the breast<br>- Cytopathology of the urinary system<br>- Cytopathology of effusions |
| [79] | 2016 | 130 | DL for digital pathology image analysis<br>- Nuclei segmentation use case<br>- Epithelium segmentation use case<br>- Tubule segmentation use case<br>- Invasive ductal carcinoma segmentation use case<br>- Lymphocyte detection use case<br>- Mitosis detection use case<br>- Lymphoma subtype classification use case |
| [80] | 2017 | 55 | Machine learning for medical imaging |
| [81] | 2017 | 58 | DL for brain magnetic resonance imaging (MRI) Segmentation: state of the art and future directions |

### 3.3. Brain and Body Machine Interfaces

The next level of the biomedical applications concerns the brain and body machine interfaces (BBMIs), which covers electrical signals generated by the brain and the muscles, acquired using appropriate sensors [5,82]. A BBMI system is composed of four main parts: a sensing device, an amplifier, a filter, and a control system [83]. For the brain interface, the system decodes and processes signals from a complex brain mechanisms to facilitate a digital interface between the brain and the computer [84]. The brain signals reflect the voluntary or involuntary neural actions generated by human's current activity. Various techniques for signal acquisition have been recently developed [85]: invasive techniques with an implantation of electrodes under the scalp (e.g., electrocorticography (ECoG)) or non-invasive techniques that do not require implanting of external objects into subject's brain. Different assessment techniques exists such as electroencephalogram (EEG), magnetoencephalography (MEG), functional magnetic resonance imaging (fMRI), and functional near infrared spectroscopy (fNIRS). After the brain–machine interface, the second part of the deep learning applied to the BMIs concerns the anomaly detection and diseases diagnosis such as identification of coronary artery disease by ECG signals [86], automated detection of myocardial infarction using ECG signals [87], electroencephalography data for seizure detection [88], and EEG diagnosis of Alzheimer's disease [89]. Some of the deep learning bibliography concerns the muscles activity, e.g. muscle–computer interface (MCI), such as electromyography movements classification for prosthetic hands [90] and gesture recognition by instantaneous surface EMG images [91]. Table 3 gives recent

survey papers on the machine learning techniques in general [92] and deep learning [93] used in the BMIs.

**Table 3.** Survey papers on deep learning in brain and body machine interfaces.

| Ref. | Year of Publi. | # of Citations | Research Topics |
|------|----------------|----------------|-----------------|
| [92] | 2018 | 19 | A review of classification algorithms for EEG based brain–computer interfaces: a 10-year update |
| [93] | 2018 | 21 | A complete survey on DL for healthcare applications based on physiological signals |
| [94] | 2006 | 85 | Machine learning in bioinformatics: a brief survey and recommendations for practitioners |

### 3.4. Public and Medical Health Management

The aim of public and medical health management (Pm-HM) is to analyze large medical data to develop and enhance health-care decisions for a well-being of humanity. Analyzing the spread of disease such as in the case of epidemics and pandemics in relation with the social behavior and the environmental factor [95] is one of the main challenge of Pm-HM in the coming years [7]. One of the main rich sources of patient information are electronic health records (EHR), which include medical history details such laboratory test results, allergies, radiology images, sensors multivariate times series (such as EEG), medications and treatment plans [7]. The analysis of such clinical information against temporal dimensions provides a valuable opportunity for deep learning in healthcare decision making [96], developing knowledge-distillation approach [97], for temporal pattern discovery over Rochester epidemiology project data [98], or to classify diagnoses given multivariate pediatric intensive care unit (PICU) time series [99]. A novel unsupervised deep feature learning method to derive a general-purpose patient representation from EHR data that facilitate clinical predictive modeling was developed by Lipton et al. [99].

Pm-HM also includes modeling lifestyle diseases, such as obesity, with relation to geographical areas. Tracking public health concerns, such as infectious intestinal diseases [100] or geographical obesity by using the social media where people's life and social interaction are publicly shared online is nowadays feasible [101]. In [102], geo-tagged images from Instagram are used to study the lifestyle diseases, such as obesity, drinking or smoking. Social media nested epidemic simulation (SimNest) via online semi-supervised DL was developed by Zhao et al. [103].

## 4. Omics

The scientific research in the Omics field is divided into two areas: DNA and Protein. In the DNA field, the most challenging tasks are the protein–DNA interactions, the gene expression prediction and the genomic sequencing. Identifying the critical genes for an early cancer detection is an open challenge [104–107]. For the protein field, most of the research is concentrated around the protein structure prediction (PSP) and the protein interaction prediction (PIP).

### 4.1. Around the Genome

Nowadays, the machine learning techniques are widely used to predict phenotypes from the genome. The usually used input features for the learning process are genomic sequences (ChIP-, RIP- and CLIP-seq). The Omics application around the genome can be divided into three areas: the protein binding prediction (PBP), the gene expression and the genomic sequencing. Figure 8 gives the whole pipeline for these two research fields.

### 4.1.1. Protein Binding Prediction

Protein–DNA interactions play important roles in different cell processes including transcription, translation, repair, and replication machinery [60]. Finding the exact location of these binding sites is very important in several domains such as drug design and development.

- **DNA–RNA-binding proteins**: The chemical interaction between the protein and a DNA or RNA strands is called the protein binding. Predicting these binding sites is very crucial in various biological activities such as gene regulation, transcription, DNA repair, and mutations [59] and it is essential for the interpretation of the genome. To model the binding sites, the position–frequency matrix is mainly used as a four input channels and the output is a binding score [108–114].
- **Enhancer and promoter identification**: Promoters and enhancers act via complex interactions across time and space in the nucleus to control when, where and at what magnitude genes are active [115]. Promoters and enhancers were early discoveries during the molecular characterization of genes [115]. Promoters specify and enable the positioning of RNA polymerase machinery at transcription initiation sites while enhancers are the short regions of DNA sequences bounded by a certain type of proteins, the transcription factors [115–119].

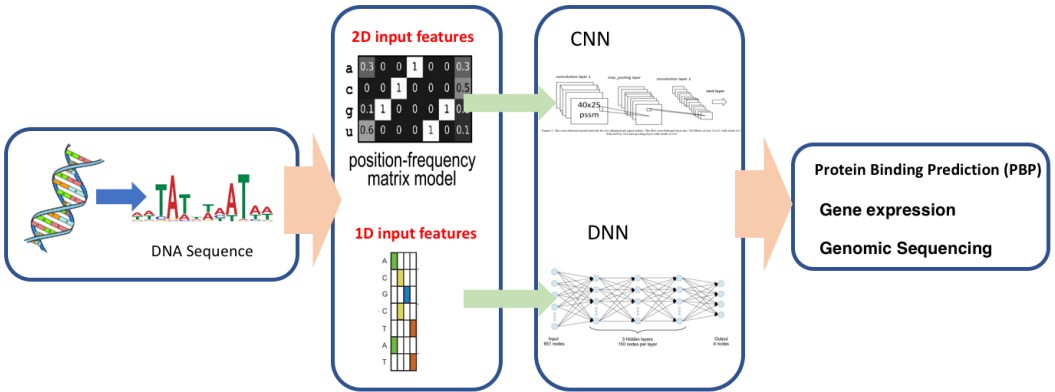

**Figure 8.** Around the genome.

### 4.1.2. Gene Expression

Gene expression is a biological process divided in three steps: the transcription, the RNA processing and the translation. The transcription creates an RNA molecule (called precursor messenger RNA (pre-mRNA)) that is essentially a copy of the DNA in the gene being transcribed. Then, RNA processing modifies the pre-mRNA into a new RNA molecule called messenger RNA(mRNA). Translation creates a protein molecule (an amino-acid chain) by reading the three-letter (codes) in the mRNA sequence [59]. The deep learning techniques are applied in gene expression according to two orientations: the alternative splicing field and the prediction of gene expression.

- **Alternative splicing**: Alternative splicing is a process whereby the exons of a primary transcript may be connected in different ways during pre-mRNA splicing [120]. The objective is to build a model that can predict the outcome of AS from sequence information in different cellular contexts [120–122].
- **Gene expression prediction**: Gene expression refers to the process of producing a protein from sequence information encoded in DNA [123]. Predicting the gene expression from histone modification can be formulated as a binary classification where the outputs represent the gene expression level (high or low) [124–126].

### 4.1.3. Genomic Sequencing

Genomic sequencing is the process of determining the precise order of nucleotides within a DNA molecule and it is nowadays very crucial in several applications such as for basic biological research, medical diagnosis, biotechnology, forensic biology, virology and biological systematics. The application of the deep learning in the genomic sequencing are divided into two fields: learning the functional activity of DNA sequencing and DNA methylation:

- **DNA sequencing**: Learning the functional activity [127], quantifying the function [128] or identifying functional effects of noncoding variants [129] of DNA sequences from genOmics data are fundamental problems in Omics applications recently touched by the enthusiasm of deep learning. De novo identification of replication domains type using replication timing profiles is also a crucial application in the genomic sequencing [130].
- **DNA methylation**: DNA methylation is a process by which methyl groups are added to the DNA molecule and can change the activity of a DNA segment without changing the sequence. DNA methylation plays a crucial role in the establishment of tissue-specific gene expression and the regulation of key biological processes [131,132].

### *4.2. Around the Protein*

The application of deep learning in the field of the protein can be divided into two areas: protein structure prediction (PSP) (Figure 9) and protein interaction prediction (PIP) (Figure 10). The commonly used features in these various protein prediction problems are [133]: physicochemical properties, protein position specific scoring matrix (PSSM), solvent accessibility, secondary structure, protein disorder, contact number and the estimated probability density function of errors (difference) between true torsion angles and predicted torsion angles based on related sequence fragments.

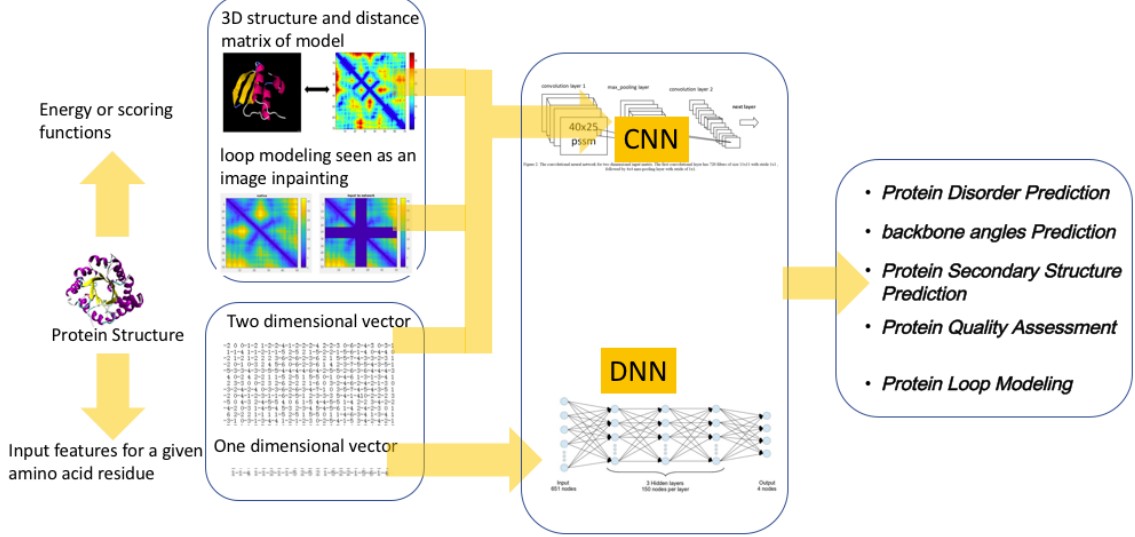

**Figure 9.** Protein Structure Prediction (PSP).

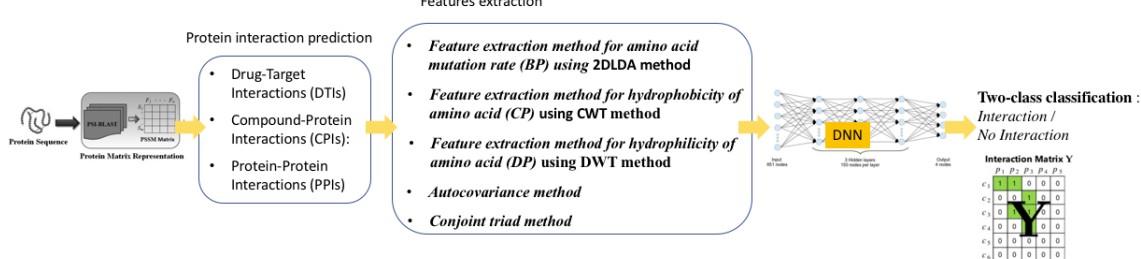

**Figure 10.** Protein Interaction Prediction (PIP).

4.2.1. Protein Structure Prediction (PSP)

Nowadays, one of the most challenging problems in Omics concerning the protein study is to obtain a better modeling and understanding of the protein folding process [58]. During the gene expression process, every protein folds into a unique three-dimensional structure that determines its specific biological function. The three-dimensional structure is also known as native conformation and it is a function of its secondary structures. Protein structure prediction is crucial to analyze protein function and applications. Several diseases are directly linked to the result of the aggregation of ill-formed proteins [134]. We highlight five main orientations in the field of the protein structure prediction. Most of these studies concern the protein secondary and tertiary structure prediction:

- **Backbone angles prediction**: Prediction of the protein backbone torsion angles (Psi and Phi) can provide important information for protein structure prediction and sequence alignment [133,135–137].
- **Protein secondary structure prediction**: Prediction of the secondary structure of the protein is an important step for the prediction of the three-dimensional structure and concentrates a large part of the scientific publications [134,138–146].
- **Protein tertiary structure (3D) prediction**: Protein tertiary structure deals with the three-dimensional protein structure and gives how the regional structures are put together in space [147–152].
- **Protein quality assessment (QA)**: In the process of protein three-dimensional structure prediction, assessing the quality of the generated models accurately is crucial. The protein structure quality assessment (QA) is an essential component in protein structure prediction and analysis [153–155].
- **Protein loop modeling and disorder prediction**: Biology and medicine have a long-standing interest in computational structure prediction and modeling of proteins. There are often missing regions or regions that need to be remodeled in protein structures. The process of predicting missing regions in a protein structure is called loop modeling [156–158]. Many proteins contain regions with unstable tertiary structure. These regions are called disorder regions [159].

4.2.2. Protein Interaction Prediction (PIP)

Most of the cellular processes involve interactions between the proteins and other proteins, drugs or compounds elements [56]. Identification of drug targets and drug target interactions are important steps in the drug-discovery pipeline and it is a great challenge in drug repositioning [57]. Identification of the interactions between chemical compounds and proteins plays a critical role in network pharmacology, drug discovery, drug target identification, elucidation of protein functions, and drug repositioning. Nowadays, different machine learning algorithms are employed for the protein interaction prediction to reveal molecular mechanisms in cellular processes. The protein interaction prediction is divided into three fields: protein–protein interactions (PPIs), drug–target interactions (DTIs) and compound–protein interactions (CPIs).

- **Protein–Protein Interactions (PPIs)**: Protein–protein interactions (PPI) play critical roles in many cellular biological processes, such as signal transduction, immune response, and cellular organization. Analysis of protein–protein interactions is very important in the drug target detection and therapy design [160–167].
- **Drug–Target Interactions (DTIs)**: Identifying drug–target interactions (DTIs) is a major challenge in drug development and provides detailed descriptions of the biological activity, genomic features and chemical structure to the disease treatment [168–170].
- **Compound–Protein Interactions (CPIs)**: The interactions between compounds and proteins plays a crucial role in pharmacology and drug discovery [171–173].

## 5. Brain and Body Machine Interface

Brain and body machine interface is defined as a combination of hardware and software that allows brain activities to control external devices or even computers [174]. Different signals with number of channels to be recorded out of the brain are acquired by different electrodes technology. These electrodes are essentially classified into two cases: invasive and non-invasive. For the invasive electrodes, neurosurgery is necessary to implant the microelectrodes in the brain under the skull while non-invasive electrodes do not need any penetration in the scalp. The signals produced by invasive electrodes are high quality, however, non-invasive electrodes are still preferable due to avoiding surgery [174]. These control signals are classified into three categories [174]: (1) evoked signals; (2) spontaneous signals; and (3) hybrid signals. Evoked signals are the signals generated unconsciously by the subject when he/she receives external stimuli. The most well-known evoked signals are steady state evoked potentials (SSEP) and P300. Spontaneous signals are the signals generated by subject voluntarily without any external stimulations [174]. Most of the well-known spontaneous signals are the motor and sensorimotor rhythms, slow cortical potentials (SCP), and non-motor cognitive tasks. Hybrid signals mean that a combination of brain generated signals are used for control.

We strongly recommend reading several complete survey papers [82,85,174–180] that provide the state of the art on brain–machine interface applications and a detail focus on definitions, classifications and comparisons of the brain signals. In addition, they survey the current brain interface hardware and software and explain the current challenges of the field.

### 5.1. Brain Decoding

#### 5.1.1. Evoked Signals

- Steady state evoked potentials (SSEP): SSEP signals are brain signals that are generated when the subject perceives periodic external stimulus [174]. Table 4 gives an overview of papers using deep learning techniques for SSEP applications. The SSEP applications are divided into three main parts: human's affective and emotion states classification, auditory evoked potential and visual evoked potential.
- P300: It is an EEG signal that appears after almost 300 ms when the subject is exposed to infrequent or surprising task [174]. Table 4 gives an overview of papers using deep learning techniques for P300 applications. The two main applications are: the P300 speller and driver fatigue classification.

#### 5.1.2. Spontaneous

- Motor and sensorimotor rhythms (MSR): Motor and sensorimotor rhythms are brain signals in relation with motor actions such as moving arms [174]. Most of the applications concerns the motor imagery tasks, which is the translation of the subject motor intention into control signals through motor imagery states (see Table 4).
- Non-motor cognitive tasks: The non-motor cognitive tasks concern all the signals generated by the brain for spontaneous music imagination, visual counting, mental rotation, and mathematical computation [174] (see Table 4).

### 5.1.3. Hybrid Signals

For a better reliability and to avoid the disadvantages of each type of signals, some techniques use the combination of several brain signals such as electroencephalography (EEG) for brain activity, electrooculography (EOG) for eye movements, and electromyography (EMG) [174,181] (see Table 4).

**Table 4.** Overview of papers using DL techniques for brain decoding applications.

| Purpose | Signal Type | NN Type | Ref. |
| --- | --- | --- | --- |
| ***Steady state evoked potentials (SSEP) applications*** | | | |
| Human's affective and emotion states classification | EEG | DBN | [182–188] |
| | | DMLP | [189,190] |
| | | CNN | [191,192] |
| Auditory evoked potential | EEG | CNN | [193–196] |
| | EMG | DMLP | [197] |
| Visual evoked potential | EEG | CNN | [198–202] |
| | | DMLP | [203] |
| | | DBN | [204–206] |
| Classification of mental load | EEG | CNN | [207] |
| ***P300 applications*** | | | |
| The P300 speller | EEG | CNN | [208–211] |
| | | DNN | [212] |
| Rapid serial visual presentation | EEG | CNN | [213] |
| Driver fatigue classification | EEG | CNN | [214–216] |
| | | DBN | [217] |
| ***Motor and sensorimotor rhythms (MSR) applications*** | | | |
| Motor imagery tasks | ECoG | CNN | [218] |
| | EEG | CNN | [219–227] |
| | | DBN | [228,229] |
| | | DMLP | [230–232] |
| ***Non-motor cognitive tasks*** | | | |
| Mental arithmetics (MA), word generation (WG) and mental rotation (MR) | fNIR | DMLP | [233] |
| Mental task | EEG | CNN | [234] |
| Mental subtractions | fNIR | DMLP | [235] |
| ***Hybrid signals*** | | | |
| Sleep stage classification | EEG, EOG, EMG | DBN | [181] |
| Emotion analysis | EEG, EOG, EMG, skin temperature, GSR, blood volume pressure, and respiration signals | DMLP | [236] |
| Detecting driving fatigue | EEG, EOG | DMLP | [237] |

### 5.2. Diseases Diagnosis

The second part of the BBMIs concerns the diseases diagnosis. Many of these applications concern the automated detection of arrhythmia and cardiac abnormalities, which are irregularities in the rate or heartbeat rhythm automatically detected by the deep neural networks. DL has also been applied in other applications such as automated detection and diagnosis of seizure, screening of depression [238] or neonatal sleep state identification [239]. Table 5 gives an overview of papers using DL techniques for diseases diagnosis and abnormalities detection.

**Table 5.** Overview of papers using deep learning techniques for Diseases Diagnosis.

| Purpose | Signal Type | NN Type | Ref. |
|---|---|---|---|
| Automated detection of arrhythmia and cardiac abnormalities | EEG | DBN | [240] |
| | | DNN | [241,242] |
| | ECG | CNN | [243–246] |
| | | DNN | [247–249] |
| | | DBN | [250–252] |
| Automated detection and diagnosis of seizure | EEG | DBN | [88] |
| | | CNN | [253–259] |
| Localization of epileptogenicity | EEG | CNN | [260] |
| Epileptiform discharge anomaly detection | EEG | DBN | [261] |
| Alzheimer's disease diagnosis | EEG | DBN | [89] |
| Neurophysiological clinical monitoring | EEG | DBN | [262] |
| Detection of hypoglycemic episodes in Children | ECG | DBN | [263] |
| Automated detection of myocardial infarction | EEG | CNN | [87] |
| Screening of depression | EEG | CNN | [238] |
| Prediction of post anoxic coma | EEG | CNN | [264] |
| Coronary artery disease (CAD) Diagnosis | EEG | CNN | [86] |
| Screening paroxysmal atrial fibrillation | ECG | CNN | [265] |
| Detection sleep apnea | ECG | DNN | [266] |
| Detecting atrial fibrillation | ECG | CNN | [267] |
| Fetal electrocardiogram (FECG) monitoring | ECG | DNN | [268] |
| Drowsiness detection | EOG | CNN | [269] |
| Neonatal Sleep State Identification | EEG | DNN | [239] |

## 6. Discussions and Outlooks

### 6.1. Overview of the Reviewed Papers

A total of 158 papers are reviewed in this paper: 64 papers for the Omics and 94 for the BBMIs. Figure 11 gives the distribution of these papers. Most of the reviewed papers (more than 50%) were published in 2017 or the first months of 2018, which shows the popularity of deep learning in the biomedical applications during the two last years. Much of the Omics research concerns the proteins, precisely protein structure prediction and protein binding prediction (see Figure 12). In the BBMI, 37% of the papers deals with the disease diagnosis and 60% concerns the brain decoding. Most of these papers use the EEG input signal (see Figure 13 for a complete overview).

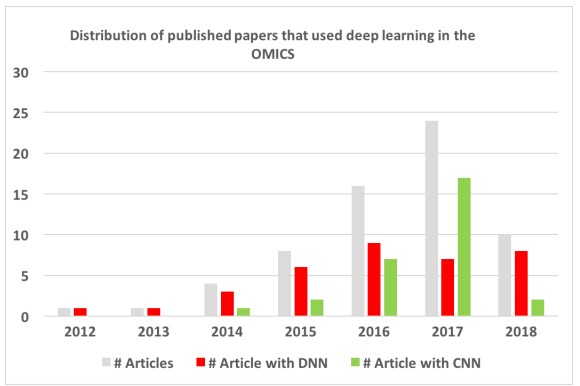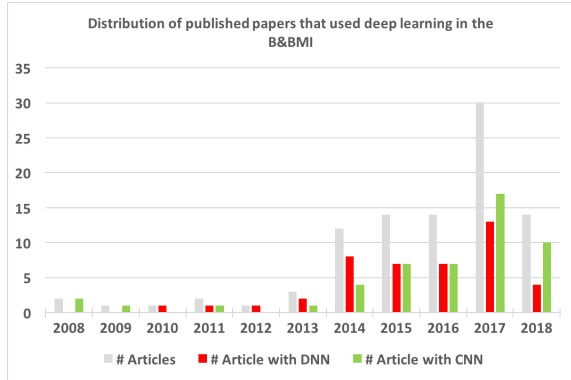

**Figure 11.** Distribution of papers reviewed in this survey.

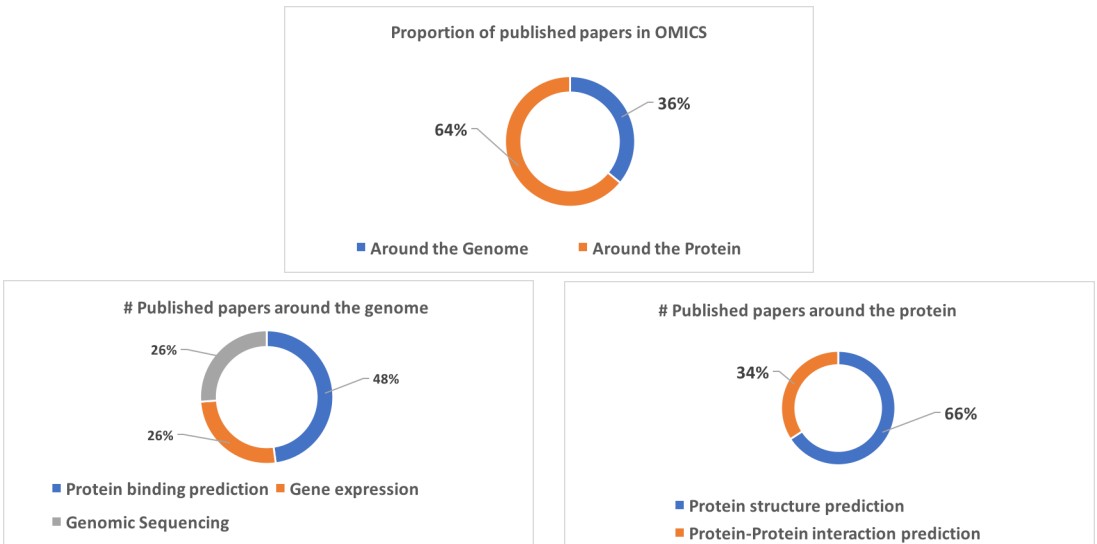

**Figure 12.** Distribution of the Omics papers reviewed in this survey.

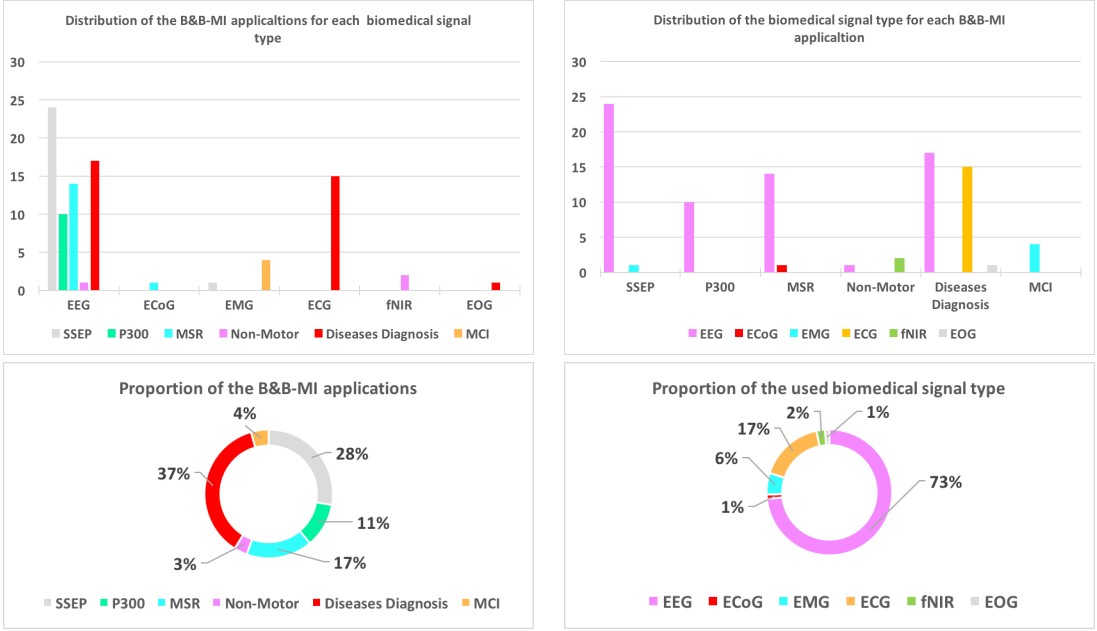

**Figure 13.** Distribution of the BBMI papers reviewed in this survey.

## 6.2. Overview of the Used Deep Neural Networks Architectures

It is well-known today that the most used deep neural networks architecture in the image analysis are the convolutional neural networks (it is also true for the medical image analysis—see the Discussion Section in [77]). We show in Figure 14 a global overview of the used deep neural architectures for the two biomedical fields (Omics and BBMI). We distinguish the two types of architectures:

- DNN: Deep neural architectures for non-locally correlated input data, which encompasses the deep multilayer perceptron, deep auto-encoders and the deep belief networks (see Section 2.2.1).
- CNN: Deep neural architectures for high locally correlated data.

Figure 11 shows that the CNNs were the preferred architecture used by researchers in the published papers during 2017. CNNs are now the most used deep neural architectures in BBMI applications, and are not far from it for the Omics (see Figure 14). However, the CNNs are not really

preferred in two Omics sub-fields: the protein–protein interaction prediction and gene expression (see Figure 14).

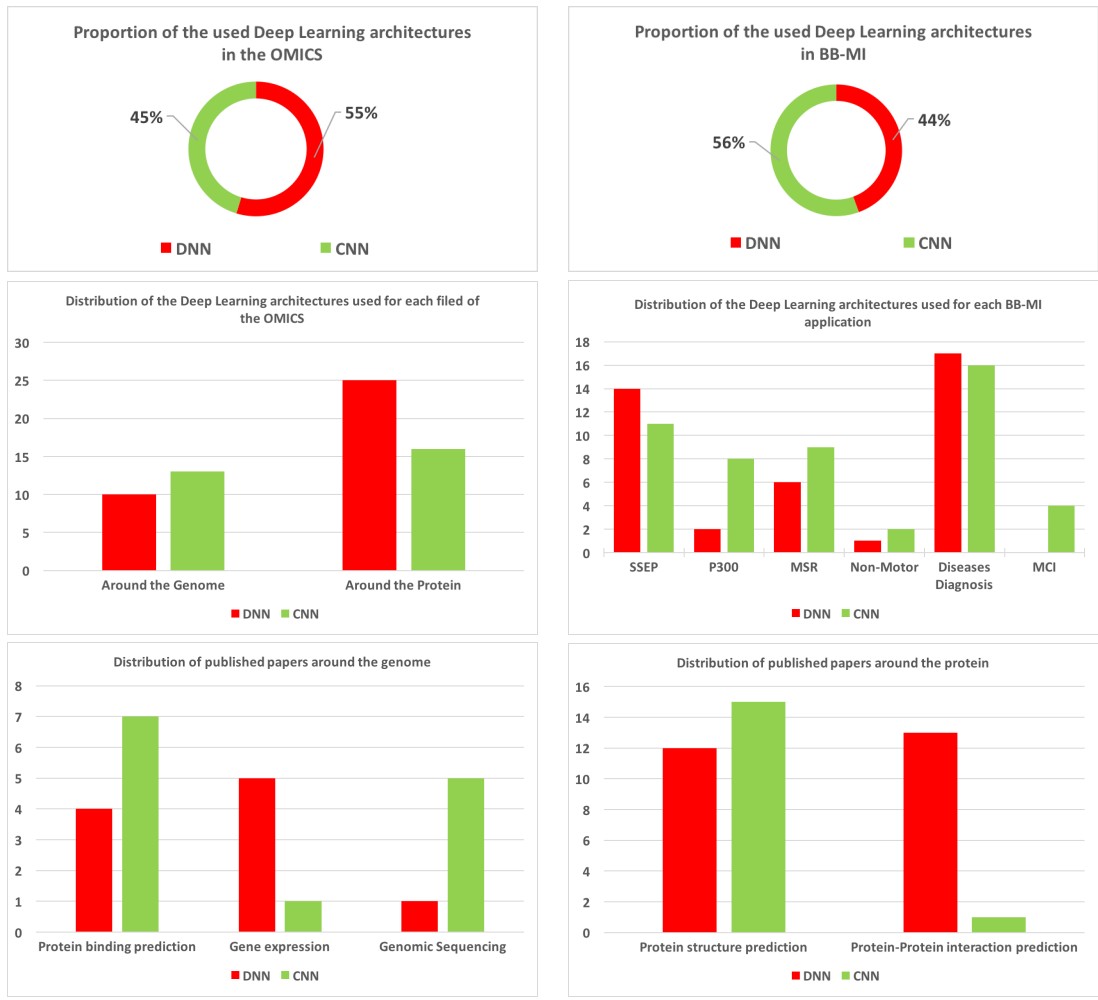

**Figure 14.** Distribution of the deep learning architectures used in the papers reviewed in this survey.

Generally, there is an increasing interest in using the CNNs in biomedical analysis and decoding. For bio and medical images analysis, we think that this interest will grow exponentially in the next few years, as well as for the other biomedical fields, such as the Omics and the BBMI. The recent publications in the image analysis and biomedical imaging show that the current standard practice is to replace the traditional handcrafted machine learning methods by the CNNs used in an end-to-end way [77]. Indeed, for the BBMI signals analysis, a complete study on how to design and train a CNN to decode task-related information from the raw EEG signals without handcrafted features is recently presented [270]. Thus, the EEG signals are represented as 2D-array-image where the number of time-step EEG signals represent the width of the image, and the number of electrodes as the height. In the Omics field, several input features could also be represented in a 2D-array, such as the position–frequency matrix model (Figure 8) or the protein structure feature (Figure 9). The dynamic sequencing data, usually tailored in the past for the recurrent neural architectures, are transformed in a 2D-array-image and easily processed by the CNNs. A great impact of this 2D representation is that the recurrent neural networks are nowadays less used in biomedical applications (see Figure 2 of [7]). This will be further accentuated by the success of some CNNs architectures such as AlexNet or GoogleNet, and also by the latest advances in Graphics Processing Units (GPUs). However, other new architectures are emerging such as DeeperBind which is a long short term recurrent convolutional network for prediction of protein binding ([109]), or convolutional deep belief networks [30].

### 6.3. Biomedical Data and Transfer Learning

Transfer learning is the ability to exploit similarities between different knowledge or dataset to facilitate the learning of a new task that shares some common characteristics. CNNs are the main deep neural architecture which have shown a great ability in transferring knowledge between apparently different image classification tasks. In most cases, the transfer learning is done by weight transferring where a network is pretrained on a source task and then the weights of some of its layers are transferred to a second network that is used for another task [271].

Recently, transfer learning was widely used in the image analysis and especially in the biomedical imaging [271–274]. Indeed, in the biomedical field, obtaining datasets that are comprehensively labeled and annotated remains a challenge (see the paragraph transfer learning and fine tuning of the guest editorial [65]). CNNs models are pre-trained from natural image dataset or from a different medical domain and then fine-tuned for new medical images. This practice is in its infancy in the Omics and BBMI. We found few papers referring to the transfer learning (e.g., [258,259] for seizure detection, [234] for mental task classification and [116] for enhancers prediction). An original transfer learning was done by Tan et al. [275], who transferred knowledge from ImageNet computer vision database to an EEG signal classification. The obtained results are very promising and will certainly open new, promising perspectives in Omics and BBMI.

### 6.4. Model Building

The different steps to follow when using an artificial neural network (shallow or deep) are: (1) model choosing; (2) model building; (3) model learning; and (4) model checking. In the first step, we must choose one neural architecture (CNN, AE, DBN, etc.). In the second step, we have to define the size of the NN: how many layers, how many units per layer, how many convolution filters and what is their size. In the third step, the neural network will be trained by unsupervised or supervised techniques while avoiding overfitting and underfitting. During the last step, we have to check the quality of the NN.

The main difficulty with the neural networks is the model building. None of the papers reviewed in this survey gives a scientific motivation about the choice of the number of hidden layers or the number of units per layer. For example, when using a CNN, it is very hard for a non-expert to define the best size of the convolution filters and to choose the best number of convolution, pooling and fully-connected layers. A good alternative to this drawback is to consider the neural architecture as a hyper-parameter evolving during the learning process. The neural network is built step by step during the learning process, until a convergence criteria is reached. To avoid an oversized architecture, some of the parameters as non-significant units or connections between neurons can be removed. Recently, several promising studies about the constructive and pruning algorithms were published (see [38,276] for a complete survey).

The other difficulty is the interpretability of the obtained results. Artificial neural networks learn to associate an output according to a given input, but they do not learn to give any reason or interpretation associated to this response. It is very hard to understand what happen in the hidden layers and why a trained NN gives a positive diagnosis for a certain pathology. This Black Box aspect is very restrictive, especially in medicine where a decision interpretability is very important and can have serious legal consequences [77]. When convolution networks are used for image processing, several methods have been developed to visualize what happen in the intermediate layers. Some of these algorithms are, for example, visual explanations from CNN networks via gradient-based localization [277]; a visualization technique of the input stimuli that excite individual feature maps at any layer in a CNN model [278]; and a deep Taylor decomposition method for interpreting generic multilayer neural networks by decomposing the network classification decision into contributions of its input elements [279]. When the input data are not images, the interpretability of the hidden layers activities is less obvious. Some visualization techniques, such as the t-distributed stochastic neighbor embedding projection (t-SNE) [280], converts a high-dimensional dataset into a 2D-matrix of

pairwise similarities. Feature maps of the model are then obtained; all the difficulty is explaining the classification decisions according to these maps.

*6.5. Emergent Architectures: the Generative Adversarial Networks*

One of the most emerging architectures used in the biomedical applications are generative adversarial networks (GANs) (Figure 15). GANs provide a way of data augmentation to enlarge the deep representations without extensively annotated training data [281]. Proposed in 2014 by Goodfellow [282], a GAN includes two deep networks: a generator and a discriminator. The first network is seen in a common analogy: an art forger who creates forgeries with the aim of making realistic images. The discriminator represents the art expert who has to distinguish between the synthetic and the authentic images [281,283]. The training of a GAN requires both finding the parameters of a discriminator and a generator, the discriminator having to maximize its classification accuracy and the generator having to confuse the discriminator as much as possible [281]. During the training process, only one of the two networks is concerned by the parameters updating. The second one keeps its own parameters frozen. Several variants of the GANs have recently been developed from the original fully-connected architecture proposed by Goodfellow [282] such as convolutional GANs, conditional GANs, adversarial autoencoders (AAE) and class experts GAN (see [281,283–286]).

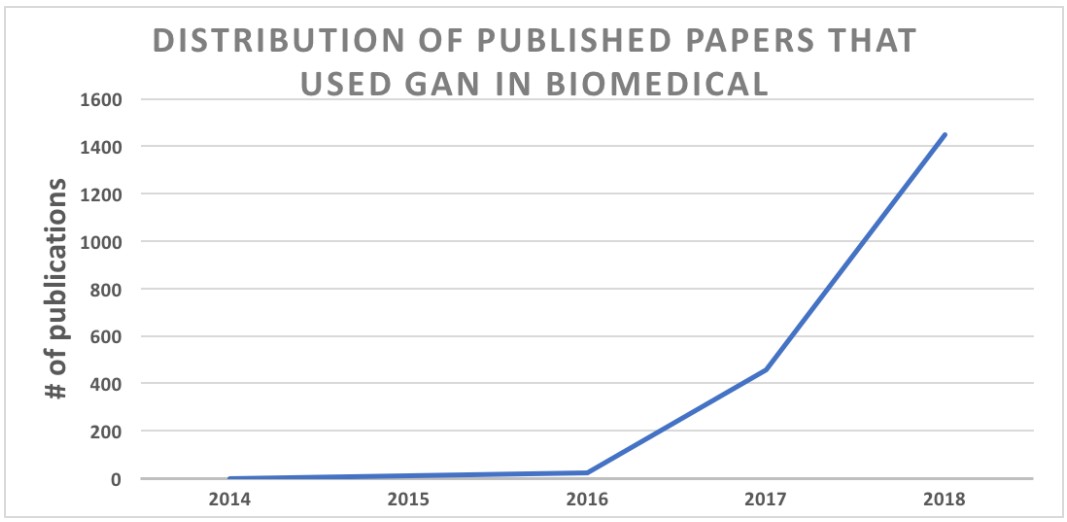

**Figure 15.** Distribution of published papers that use Generative Adversarial Networks (GANs) in biomedical applications. The statistics are obtained from Google Scholar.

GANs are recently applied in all biomedical fields such as in the Omics for a protein modeling by Li et al. [156], who considered loop modeling as an image inpainting problem with the generative network having to capture the context of the loop region with a prediction of the missing area.

In the BBMI applications, such as cardiac ECG Applications, a novel concept that embeds a generative variational autoencoder (VAE) into the objective function of Bayesian optimization was applied to estimating tissue excitability in a cardiac electrophysiological model by Dhamala et al. [287]. In [288], a deep generative model is trained to generate the spatiotemporal dynamics of transmembrane potential (TMP).

In bioimaging histology applications, to classify the newly given prostate datasets into low and high Gleason grade, an adversarial training is used to minimize the distribution discrepancy at the feature space, with the loss function adopted from the GAN [289]. In [290], a cascaded of refinement GANs for phase contrast microscopy image super-resolution is proposed.

Most research using GANs concerns medical imaging applications. The main objectives are: image quality enhancement, image reconstruction, crafted images generation, image registration and segmentation.

The first objective of the GANs in medical imaging is to enhance the image quality. A conditional GAN is used for the reduction of metal artifacts (RMA) in computed tomography (CT) ear images of cochlear implants (CIs) recipients in [291], and to estimate the high-quality full-dose PET images from low-dose ones in [292]. To reduce the artifacts from sparsely reconstructed cone-beam CT (CBCT) images, Liao et al. [293] introduced a least squares generative adversarial networks (LSGAN), which is formulated under an image-to-image generative model. Chen et al. [294] introduced a 3D neural network design, namely a multi-level densely connected super-resolution network (mDCSRN) with a GAN to generate a high-resolution (HR) magnetic resonance images (MRI) from one single low-resolution (LR) input image. Similarly, a deep residual network (ResNet) is used as a part of a GAN and trained to enhance the quality of ultrasound images [295]. In [296], a noise reducing generator based on a CNN together with a GAN is developed.

In image reconstruction methods, 3D anatomical structures remain a challenge when limited number of 2D views are used to reduce cost and radiation exposure to patients. A new deep conditional generative network for the 3D reconstruction of the fetal skull from 2D ultrasound (US) standard planes of the head routinely acquired during the fetal screening process was introduced by Cerrolaza et al. [297]. This 3D reconstruction is based on the generative properties of a conditional variational autoencoders (CVAE), a deep conditional generative network for the 3D reconstruction of the fetal skull from freehand non-aligned 2D scans of the head. In another part, compressed sensing-based magnetic resonance imaging (CS-MRI) is a promising paradigm allowing to accelerate MRI acquisition by reconstructing images from only a fraction of the normally required measurements [298]. To solve this problem, deep generative adversarial networks were recently used by Seitzer et al. [298], Zhang et al. [299], and Quan et al. [300].

Due to the lack of annotated data, GANs are used to generate crafted images, for example to synthesize multiple realistic-looking retinal images from an unseen tubular structured annotation that contains the binary vessel morphology in [301] or to generate synthetic liver lesion images in [302]. In [303], adversarial examples are explored in medical imaging and leveraged in a constructive fashion to benchmark model performance not only on clean and noisy but also on adversarially crafted data. In [304], a conditional GAN is explored to augment artificially generated lung nodules to improve the robustness of the progressive holistically nested network (P-HNN) model for pathological lung segmentation of CT scans. In [305], a novel generative invertible networks (GIN), which is a combination of a convolutional neural network and generative adversarial networks, is proposed to extract the features from real patients and generate virtual patients, which are both visually and pathophysiologically plausible, using the features. In [306], a deep generative multi-task is developed to solve the problem of limited training data and data with lesion annotations because making the annotations is a very expensive and time consuming task. The deep generative multi-task is built upon the combination of a convolutional neural networks and a generative adversarial network. Transfer learning is also adapted to a model trained with one type of data to another. In [307], a conditional generative adversarial network is used to generate realistic chest X-ray images with different disease characteristics by conditioning its generation on a real image sample. This approach has the advantage of overcoming limitations of small training datasets by generating truly informative samples. To solve the problem of missing data in multi-modal studies, Pan et al. [308] proposed a two-stage deep learning framework for Alzheimer's disease diagnosis using incomplete multi-modal imaging data. In the first stage, a 3D cycle consistent generative adversarial networks model is used for imputing the missing PET data based on their corresponding MRI data. A landmark-based multi-modal multi-instance neural network for brain disease classification is used in the second stage.

Image registration can be defined as the process of aligning two or more images to find the optimal transformation that best aligns the structures of interest in the input images. Fan et al. [309] introduced an adversarial similarity network, which is inspired by generative adversarial network, to automatically learn the similarity metric for training a deformable registration network. In [310], an adversarial learning approach is used to constrain convolutional neural network training for image

registration. The end-to-end trained network enables efficient and fully-automated registration that only requires a magnetic resonance and transrectal ultrasound image pair as input, without anatomical labels or simulated data during inference.

Generative adversarial networks are also used in medical imaging segmentation. In [311], a novel real-time voxel-to-voxel conditional generative adversarial nets is used for 3D left ventricle segmentation on 3D echocardiography. For automatic bony structures segmentation, a cascaded generative adversarial network with deep-supervision discriminators was used by Zhao et al. [312]. In [313], a novel transfer-learning framework using generative adversarial networks is proposed for robust segmentation of different HEp-2 datasets. A recurrent generative adversarial network called Spine-GAN to perform automated segmentation and classification of intervertebral discs, vertebrae, and neural foramen in MRIs in one shot was developed by Han et al. [314]. Jiang et al. [315] aimed to solve the problem of learning to segment for lung cancer tumors from MR images through domain adaptation from CT to MRI, where there are a reasonable number of labeled data in the source domain CT but are provided with very limited number of target samples in MRI with fewer labels. For breast mass segmentation in mammography, conditional generative adversarial and convolutional networks was used by Singh et al. [316]. For automatic segmentation and classification of images from patients with cardiac diseases associated with structural remodeling, a variational autoencoder (VAE) model based on 3D convolutional layers was introduced by Biffi et al. [317]. A multitask generative adversarial networks was proposed by Xu et al. [318] as a contrast-free, stable and automatic clinical tool to segment and quantify myocardial infarction simultaneously. In [319], a task driven generative adversarial network is introduced to achieve simultaneous image synthesis and automatic multi-organ segmentation on X-ray images.

To model the respiratory motion for tracking mobile tumors in the thorax and abdomen in a radiotherapy, Giger et al. [320] developed a conditional generative adversarial network trained to learn the relation between temporally related ultrasound and 4D MRI navigator images.

## 7. Conclusions

Deep neural networks are currently the main machine learning tools in all areas of the biomedical applications, such as in the Omics, bio and medical imaging, brain and body machine interface and public health management. We give in this survey paper the recent status of the deep learning in the biomedical applications. Among all the deep neural architectures, there is a growing interest in an end-to-end convolutional neural network, replacing the traditional handcrafted machine learning methods. The analysis of the literature shows that CNNs are the main deep neural architecture and have shown a great ability in transferring knowledge between apparently different classification tasks, done by weight transferring. In most cases, the transfer learning concerns the biomedical imaging applications.

The main emergent architectures used in the biomedical applications are the generative adversarial networks. These amazing neural architectures provide a data augmentation method to enlarge the deep representations, without extensively annotated training data. The GANs are mainly applied in biomedical imaging applications.

Despite the great success of deep neural networks in biomedical applications, many difficulties such as model building or the interpretability of the obtained results are encountered by deep learning users.

**Author Contributions:** These authors contributed equally to this work.

**Funding:** This research received no external funding.

**Conflicts of Interest:** The authors declare no conflict of interest.

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
