# Peer review of "Deep Learning in the Biomedical Applications: Recent and Future Status"

_applsci, doi:10.3390/app9081526_

Round 1

Reviewer 1 Report

Suggestions about text organization:

- Check the contents for sections 3,4 and 5 for avoid repeating ideas. 

About style and format:

- Review on the English language to avoid some minor mistakes. 

- Using cursive font rather than capitals to emphasize text and improve readability. 

- In general all the figures should be revised to improve their legibility specially with proper size choice and font. 

- Emphasize of the heading row in all tables. Also, the horizontal alignment of text for each cell should be revised. Note a missing reference in table 5.

Author Response

Response to Reviewer 1 Comments

Dear reviewer, we address your comments below. Our responses are in red italics and embedded after the questions.

1. Check the contents for sections 3,4 and 5 for avoid repeating ideas.

Section 3,4 and 5 were checked, some repeating ideas were removed

2. Review on the English language to avoid some minor mistakes.

We have read carefully all the manuscript

3. Using cursive font rather than capitals to emphasize text and improve readability.

We removed all the unnecessary capital letters from the manuscript.

4. In general all the figures should be revised to improve their legibility specially with proper size choice and font.

We have checked all the figures for a better legibility.

5. Emphasize of the heading row in all tables. Also, the horizontal alignment of text for each cell should be revised. Note a missing reference in table 5.

All the table were revised.

The missing reference in table 5 is corrected

Best Regards

Ryad Zemouri

Cedric Lab /Cnam

[email protected]

Reviewer 2 Report

The manuscript authored by Zemouri et al. gave a complete survey of the deep learning application in biomedical applications including omics, medical imaging, brain/body machine interface, and health management. The topic is very important and they have provided enough references for the readers. However, the manuscript suffers from several weaknesses in presentation and format. It is necessary to have someone else proof-read the paper. Here are some comments:

1.          It is rare to see a typo in the title: Deep learning in the biomedical applications: Recent and “future” status

2.          Too many unnecessary capital letters are in the manuscript. For example, image “R”econstruction.

3.          Abbreviations should be indicated once at their first appearance through the manuscript. I saw generative adversarial network (GAN) and convolutional neural network (CNN) again and again.

4.          The authors gave further descriptions for Omics and Brain and Body Machine Interface in section 4 and 5, respectively. I understand the authors did not discuss the health management since that is a little far from conventional biomedical application. But how about medical imaging? This is also one of the main topics as mentioned by the authors in the Abstract and section 3. There are many machine learning works in the field of medical imaging. I suggest to add a section for that.

5.          In abstract, “…the BB-MI as Public and Medical Health Management (PmHM)”. What does that mean?

6.          In Fig 1, the author only used keyword "deep learning" in their searching. I suggest the similar terms should also be included. For example, CNN or DNN.

7.          In Fig. 4, W2 should be Wn. "*" is not the proper math symbol.

8.          Please make sure the term "statics NNs" is correct.

9.          What does Fig. 7C mean? Please give further information. Why are there no connections between nodes in the input layers and the 4th node in the hidden layer in Fig 7B and C?

10.      There is a missing reference [?] in Table 5.

11.      In line 519, the author mentioned "they do not learn to give any reason or interpretation associated to this response". However, the interpretation could be made based on the feature maps of the model. Also, I would like to see how the black box can be improved in this section.

12.      I suggest the authors to rewrite the conclusions. The contents are redundant with those in the previous sections. For example, the sentences in line 656 and line 513 are almost same.

13.      In line 409, “We strongly recommend to ‘reed’ these complete survey papers…”

14.      I would suggest to add this reference to survey papers about BCI in line 409: “Challenges and Future Perspectives on Electroencephalogram-based Biometrics in Person Recognition” Frontiers in Neuroinformatics.

Author Response

Response to Reviewer 2 Comments

Dear reviewer, we address your comments below. Our responses are in red italics and embedded after the questions.

The manuscript authored by Zemouri et al. gave a complete survey of the deep learning application in biomedical applications including omics, medical imaging, brain/body machine interface, and health management. The topic is very important and they have provided enough references for the readers. However, the manuscript suffers from several weaknesses in presentation and format. It is necessary to have someone else proof-read the paper. Here are some comments:

Thank you for these comments. We have read carefully all the manuscript

1.          It is rare to see a typo in the title: Deep learning in the biomedical applications: Recent and “future” status

This typo is corrected.

2.          Too many unnecessary capital letters are in the manuscript. For example, image “R”econstruction.

We removed all the unnecessary capital letters from the manuscript.

3.          Abbreviations should be indicated once at their first appearance through the manuscript. I saw generative adversarial network (GAN) and convolutional neural network (CNN) again and again.

We reviewed the use of all the abbreviations for the whole manuscript.

4.          The authors gave further descriptions for Omics and Brain and Body Machine Interface in section 4 and 5, respectively. I understand the authors did not discuss the health management since that is a little far from conventional biomedical application. But how about medical imaging? This is also one of the main topics as mentioned by the authors in the Abstract and section 3. There are many machine learning works in the field of medical imaging. I suggest to add a section for that.

Exactly, you are right. Given the number of survey papers on medical imaging, we did not want to develop it. For lack of time (we had 10 days to submit an improved version of the manuscript) we did not have time to develop this section.

5.          In abstract, “…the BB-MI as Public and Medical Health Management (PmHM)”. What does that mean?

Sorry for this typo. The corrected sentence is :

« …. the BBMI (study of the brain and body machine interface) « and finally » the public and medical health management (PmHM). »

6.          In Fig 1, the author only used keyword "deep learning" in their searching. I suggest the similar terms should also be included. For example, CNN or DNN.

Yes, I agree with you, but if we use several keywords we will have redundancies in the results. The same article can be counted several times. This graphic gives a "non-exhaustive" evolution of scientific papers dealing with deep learning.

7.          In Fig. 4, W2 should be Wn. "*" is not the proper math symbol.

Nice catch. It’s corrected

8.          Please make sure the term "statics NNs" is correct.

You're right, it's a bad translation from French into English. We replace the term « static » by  feedforward .

9.          What does Fig. 7C mean? Please give further information. Why are there no connections between nodes in the input layers and the 4th node in the hidden layer in Fig 7B and C?

We have checked this figure and give more details about each neural architecture.

10.      There is a missing reference [?] in Table 5.

It’s corrected ,

11.      In line 519, the author mentioned "they do not learn to give any reason or interpretation associated to this response". However, the interpretation could be made based on the feature maps of the model. Also, I would like to see how the black box can be improved in this section.

We have more developed this section to address your comments. The added paragraphs are :

« A good alternative to this drawback is to consider ….. ….. the constructive and pruning algorithms were published. »

« When the convolution networks are used….. ….. the classification decisions according to these maps. »

12.      I suggest the authors to rewrite the conclusions. The contents are redundant with those in the previous sections. For example, the sentences in line 656 and line 513 are almost same.

We have rewrite a more concise conclusion to avoid redundancies.

13.      In line 409, “We strongly recommend to ‘reed’ these complete survey papers…”

Sorry for this typo. It’s corrected ,

14.      I would suggest to add this reference to survey papers about BCI in line 409: “Challenges and Future Perspectives on Electroencephalogram-based Biometrics in Person Recognition” Frontiers in Neuroinformatics.

Done, the reference was added.

Best Regards

Ryad Zemouri

Cedric Lab /Cnam

[email protected]

Reviewer 3 Report

This review is very comprehensive and well-organized. It covers the main feild of biomedical research such as OMICS, BioImaging, BB-MI, PmHM. The basic knowledge about the deep learning and neural network can help the readers is very concise. The applications are listed clearly. The future directions of deep learning have been apointed out in the Section 6.

I only have several minor comments:

(1) Figure 8. "AroundTheGenome" should be "Around The Genome".

(2) Line 466 and 468, the authors mentioned figure.14.a, .b, 14.e, .f. I think in Figure 14, a, b, c, d, e, f should be given.

(3) In 6.4, I suggest the authors give some work about interpretability of CNN because there are some great work that had been done to explain why CNN works. For example, <<Visualizing and Understanding Convolutional Networks>>, <<Grad-CAM:Visual Explanations from Deep Networks via Gradient-based Localizatio>>.

(4) The font of the words in Figure 8, 9, 10, is too small to read. Please check all the figures.

(5) Line 219, "Photon" should be "Positron".

Author Response

Response to Reviewer 3 Comments

Dear reviewer, we address your comments below. Our responses are in red italics and embedded after the questions.

This review is very comprehensive and well-organized. It covers the main feild of biomedical research such as OMICS, BioImaging, BB-MI, PmHM. The basic knowledge about the deep learning and neural network can help the readers is very concise. The applications are listed clearly. The future directions of deep learning have been apointed out in the Section 6.

Thank you for these comments.

(1) Figure 8. "AroundTheGenome" should be "Around The Genome".

Nice catch. It’s corrected

(2) Line 466 and 468, the authors mentioned figure.14.a, .b, 14.e, .f. I think in Figure 14, a, b, c, d, e, f should be given.

You're right, we have just mentioned figure 14.

(3) In 6.4, I suggest the authors give some work about interpretability of CNN because there are some great work that had been done to explain why CNN works. For example, <<Visualizing and Understanding Convolutional Networks>>, <<Grad-CAM:Visual Explanations from Deep Networks via Gradient-based Localizatio>>.

Thank you for these advices. We have more developed this section to address your comments. The added paragraphs are :

« A good alternative to this drawback is to consider ….. ….. the constructive and pruning algorithms were published. »

« When the convolution networks are used….. ….. the classification decisions according to these maps. »

(4) The font of the words in Figure 8, 9, 10, is too small to read. Please check all the figures.

We have checked all the figures for a better legibility.

(5) Line 219, "Photon" should be "Positron".

Nice catch. It’s corrected

Best Regards

Ryad Zemouri

Cedric Lab /Cnam

[email protected]

Round 2

Reviewer 1 Report

It is truly appreciate the changes done to the paper.

A few typos (misspellings) remains:

line 197: Biomedecine (fr) -> Biomedicine (en)

lined 236: bimedical -> biomedical

line 276: filed -> field

Author Response

Response to Reviewer 1 Comments

Dear reviewer, we address your comments below. Our responses are in red italics and embedded after the questions.

It is truly appreciate the changes done to the paper.

Thank you.

A few typos (misspellings) remains:

line 197: Biomedecine (fr) -> Biomedicine (en)

It has been corrected.

lined 236: bimedical -> biomedical

It has been corrected.

line 276: filed -> field

It has been corrected.

Best Regards

Ryad Zemouri

Cedric Lab /Cnam

[email protected]

Reviewer 2 Report

The authors have addressed my comments and the manuscript has much improved. Some minor comments remain and I strongly suggest the authors to carefully check the manuscript again:

In lin 144: Auto-Encoders (AEs) has already defined. 

What does "pssm" mean in Fig. 7(H). I found the description in section 4.2, I suggest to give it a definition or remove it in Fig. 7.

In line 269: remove parentheses 

"Alternative splicing (AS)" appear three times in page 11

In line 442: "firsts"

In line 642: "G"enerative "A"dversarial "N"etworks

Author Response

Response to Reviewer 2 Comments

Dear reviewer, we address your comments below. Our responses are in red italics and embedded after the questions.

The authors have addressed my comments and the manuscript has much improved. Some minor comments remain and I strongly suggest the authors to carefully check the manuscript again:

In lin 144: Auto-Encoders (AEs) has already defined.

It has been corrected.

What does "pssm" mean in Fig. 7(H). I found the description in section 4.2, I suggest to give it a definition or remove it in Fig. 7.

We changed the representation of the CNN by a more appropriate image.

In line 269: remove parentheses

It has been corrected.

"Alternative splicing (AS)" appear three times in page 11

It has been corrected.

In line 442: "firsts"

It has been corrected.

In line 642: "G"enerative "A"dversarial "N"etworks

It has been corrected.

Best Regards

Ryad Zemouri

Cedric Lab /Cnam

[email protected]
